# Measurement of Tear Production and Intraocular Pressure in Clinically Conscious Normal Captive Red Deer (*Cervus elaphus*)

**DOI:** 10.3390/ani14060940

**Published:** 2024-03-19

**Authors:** Liga Kovalcuka, Aija Malniece

**Affiliations:** Clinical Institute, Faculty of Veterinary Medicine, Latvia University of Life Sciences and Technologies, K. Helmaņa iela 8, LV-3004 Jelgava, Latvia; aija.malniece@lbtu.lv

**Keywords:** tear production, intraocular pressure, red deer, *Cervus elaphus*

## Abstract

**Simple Summary:**

Normal values of tear production (STT) and intraocular pressure (IOP) are important values resulting from ophthalmic examinations in animals, but so far there are no data from conscious red deer. The establishment of reference value ranges specific to conscious animals may be useful to institutions in settings where chemical immobilization is not necessary for the examination of the species. A total of 60 red deer were examined in this study. Both eyes of all red deer underwent a complete ophthalmic examination, including evaluation of tear production with the Schirmer tear test and IOP readings. The mean ± standard deviation of STT values were 18.35 ± 4.54 mm/30 s in the right eye and 17.87 ± 4.12 mm/30 s in the left; for both eyes, 18.11 ± 4.33 mm/30 s, with a reference range of 17.33–18.89 mm/30 s. IOP was as follows: 26.96 ± 4.42 mmHg in the right eye and 26.67 ± 3.80 mmHg in the left; for both eyes, 26.81 ± 4.11 mmHg, with a reference range of 26.07–27.55 mm/30 s. No statistically significant differences were found between the right and left eye in terms of IOP and STT values. This study provides the normal range for STT and IOP in healthy red deer. It demonstrates that STT is a practical method for determining tear production, and rebound tonometry is a practical method for evaluating IOP in ophthalmic examinations of deer.

**Abstract:**

Normal values of tear production (STT) and intraocular pressure (IOP) have not been reported in conscious red deer to date. The objective was to determine the normal range of STT and IOP in clinically healthy, conscious red deer (*Cervus elaphus*) by utilizing a chute restraint. A total of 60 red deer were examined in this study. Both eyes of all red deer underwent a complete ophthalmic examination, including evaluation of tear production with the Schirmer tear test (STT) and assessment of intraocular pressure (IOP) with rebound tonometry, employing the TonoVet^®^ device. The mean ± standard deviation of STT values were 18.35 ± 4.54 mm/30 s in the right eye and 17.87 ± 4.12 mm/30 s in the left; for both eyes, 18.11 ± 4.33 mm/30 s, with a reference range of 17.33–18.89 mm/30 s. IOP was as follows: 26.96 ± 4.42 mmHg in the right eye and 26.67 ± 3.80 mmHg in the left; for both eyes, it was 26.81 ± 4.11 mmHg, with a reference range of 26.07–27.55 mm/30 s. No statistically significant differences were observed between the IOP and STT values of the right and left eyes. This study provides a reference range for the STT and IOP in healthy red deer, showing that STT determination of tear production, and rebound tonometry to evaluate the IOP methods, are practical methods for ophthalmic examination in deer.

## 1. Introduction

The red deer (*Cervus elaphus*) belongs to the mammalian class, Artiodacytla order, and Cervidae family [1], and is one of the largest and most populous herbivore species (population of ~69,000 red deer in 2023), after roe deer (*Capreolus capreolus*, population of ~230,000 in 2023), in Latvia [2]. The red deer is native to continental Europe, apart from Northern Scandinavia, Finland, and Iceland [3].

These days, captive red deer are also kept on deer farms, where animals live in a complex field and forest environment and are mainly used for meat production, breeding, tourism, biological grass processing, and trophy hunting game. Latvia recently registered around 13,846 captive red deer [4]. 

Interest in captive animal medicine has increased due to the increase in deer farms and rising wild populations. Deer are susceptible to a variety of ophthalmic conditions such as congenital, acquired, and infectious diseases that probably increase with age. To perform adequate ophthalmic examinations and make definitive diagnoses, relevant ocular parameters must be measured. Thus, basic diagnostic tests such as the Schirmer tear test (STT) and intraocular pressure (IOP) are essential [5,6]. 

Adequate tear production, measured with STT, is important to maintain the health of the ocular surface. Tears are also essential to remove foreign matter, provide nutrients to the avascular cornea, and provide defence of the cornea through immunoglobulins, lysozymes, and other proteins [6,7]. Decreased production of tears can cause keratoconjunctivitis sicca (KCS), which can lead to ocular discomfort, pain, severe inflammation of the cornea and conjunctiva, and even the perforation of the cornea and vision loss [6,7]. 

Intraocular pressure is a balance between aqueous humour production and outflow. Increased IOP is characteristic of glaucoma, while decreased IOP is a clinical sign of intraocular inflammation [6]. On the other hand, glaucoma in small ruminants is also associated with the presence of ocular inflammatory conditions such as severe keratoconjunctivitis, corneal ulcers, anterior uveitis, ocular trauma, and septicaemia, as a result of possible extensive anterior synechiae or filtration angle obstruction with inflammatory cells or fibrin, leading to secondary elevated IOP and potential glaucoma formation [7].

Many other ocular diseases cause discomfort and therefore probably reduce production, cause economic losses, and also decrease the quality of life and welfare of captive animals. 

Thus far, there have been limited studies giving the normal values of STT and IOP in wild ruminants. There are data on the captive mouflon, Alpine ibex, Alpine chamois, captive eland, Nubian ibex [8,9,10], fallow deer, and white-tailed deer [11,12]. However, there are differences in methodology between these studies: For example, examining animals under anaesthesia or physical restraint; or measuring IOP with different tonometers (applanation or rebound) [13,14,15]. Moreover, the difference in IOP and STT values between anaesthetised and non-anaesthetised wild animals is not known, but in horses and dogs, there are many publications showing the significant influence of anaesthetic drugs on IOP and STT values [16,17]. Modern trends in veterinary medicine and zoology include animal examination without general anaesthesia, or with less usage of it, for nonpainful procedures.

To our knowledge, the Schirmer tear test and intraocular pressure have not been reported in conscious red deer restrained in a chute system, which, in wild animals, is considered a more humane way of shortening the examination time. The aim of this study was to report intraocular pressure values using rebound tonometry, and measure tear production over 30 s, in a group of healthy, conscious, non-anaesthetised female red deer by utilizing a chute restraint. Establishing normal ranges specific to conscious animals can be useful for institutions that do not use chemical immobilization to examine the species.

## 2. Materials and Methods

The field study was performed with full respect for the ethical criteria and welfare of the deer involved. The study proposal was reviewed and approved by the ethics committee of the Latvian University of Life Sciences and Technology (Nr. LLU_DZAEP_2019/11, 16.05.2019). All animals were living in outdoor enclosures with grass, bushes, and trees. Their daily diet consisted of mixed grass and hay fed ad libitum and adjusted according to the season and the species’ requirements. This study was conducted during a routine annual clinical evaluation in the spring, as part of a vaccination and deworming programme administered by the farm authority. Deer were restrained in specially designed hydraulic soft-wall automatic “crush” equipment, immobilising the doe’s body but putting no pressure on the neck.

In field conditions, 100 randomly selected red deer were sent through the hydraulic soft-wall automatic “crush” equipment, and ophthalmic examination was performed. Forty deer with indications of ophthalmic diseases that can lead to IOP and STT variations were excluded from the study. A total of 60 healthy adult red deer (so 120 eyes) were included in this study. The age of the deer varied between 3 and 11 years.

The routine ocular examination included ophthalmic examination from a distance, followed by close-up examination with direct illumination, biomicroscopy (Kowa SL15, Nagoya, Aichi, Japan), and monocular ophthalmoscopy with a PanOptic^®^ ophthalmoscope (Welch Alynn, Romford, UK). All deer included in this study were examined to ensure that they were ophthalmologically healthy.

To measure tear production, standardised sterile Schirmer tear test I (STT-I) strips (Eickemeyer, Tuttlingen, Germany) were used in both eyes (Table 1). The tip of the strip was inserted in the eye, under the lower lateral eyelid margin in the conjunctival fornix, for 30 s. After the removal of the test strip, the length of the wet area of the strip was immediately measured in millimetres.

All tonometric measurements were performed by the same person, employing rebound tonometry with a TonoVet^®^ tonometer (TonoVet^®^, Tiolat Ltd., Helsinki, Finland) on the horse (H) calibration setting, with automatic calibration provided by the device. Each measurement recorded was the automatically generated average after five successive readings (Table 1). The use of topical anaesthesia is not required for this type of tonometer, which is more practical for captive animals and beneficial to the animals due to reported corneal endothelial and systemic toxicity during frequent use of topical anaesthetics [18,19]. A single-use probe was used for TonoVet^®^ and held perpendicular to the central cornea, approximately 4 mm from the surface. No compression of the jugular veins or cervical region was applied during measurement.

All measurements were performed at approximately the same time of the day (9.00–15.00) because of the possible diurnal variation in tear production and intraocular pressure [20,21], and by the same examiner in the same body position to minimize variations in restraint or study technique. Animals showing high stress or agitation were excluded from the study.

### Statistical Method

A statistical analysis of the data was performed using the statistical software programs SPSS (Version 27.0; IBM Corp©, Armonk, NY, USA) and Microsoft Excel (Version 16.80, 2021, Microsoft Corp., Redmond, WA, USA). The arithmetic mean values (X), expressed as mean ± standard deviation (SD), and the normal range of STT and IOP were expressed for each eye separately and both eyes together. The normal range for STT and IOP was estimated by calculating the confidence interval, i.e., the lower and upper limits of the means with 95% confidence, using SPSS software.

A paired sample *t*-test was used to compare the STT and IOP values obtained from the right and left eyes. Statistical significance was defined as a *p*-value < 0.05.

## 3. Results

Thirty-nine deer were excluded from the study due to ophthalmic diseases that can lead to IOP and STT variations. The main reasons for exclusion were: Imperforated lacrimal punctum (2), cataract (2), corneal ulcer (4), healed corneal or globe trauma of unknown cause (5), conjunctivitis with discharge, and mild blepharospasms (26). One deer was excluded because of extreme behavioural symptoms of stress. Finally, 60 deer were evaluated as clinically and ophthalmologically healthy, and their STT and IOP values were used for further calculations. Descriptive statistics and suggested reference intervals for STT and IOP measurements in the deer are summarised in Table 1.

The mean ± standard deviation of STT values was 18.35 ± 4.54 mm/30 s in the right eye and 17.87 ± 4.12 mm/30 s in the left eye; there was no significant difference between the right and left eyes in terms of STT (*p* > 0.05). STT in both eyes was calculated as 18.11 ± 4.33 mm/30 s, with a reference range of 17.33–18.89 mm/30 s.

IOP in the right eye was 26.96 ± 4.42 mmHg and in the left eye 26.67 ± 3.80 mmHg. No statistically significant differences were found between the right and left eye in terms of IOP or STT values (*p* > 0.05). The IOP in both eyes was 26.81 ± 4.11 mmHg, with a reference range of 26.07–27.55 mmHg (Table 1).

The values of STT and IOP in different age groups are shown in Table 2. Groups were not compared because of the different animal group size.

## 4. Discussion

The STT strips were placed without difficulty, and deer tolerated the STT tests and IOP measurements well. To decrease examination time and simplify eye examination in wild animals, STT tests are for 30 s instead of for one minute, as they are in dogs and cats [22,23]. In dogs and horses, it has been shown that an STT of 30 s allows researchers to obtain an accurate diagnosis of tear production compared with the standard 60 s value [24]. Therefore, taking into account the purpose of decreasing the restraint time, and for safety reasons, we measured STT for 30 s [25,26]. The possibility of testing conscious, non-anaesthetised animals, and the fact that the proposed method is shorter, may make it a suitable alternative to the 1 min test [24].

Despite STT and IOP values being published for many herbivores and cervids [9,10], to the authors’ knowledge, there are no data on normal values of STT and IOP in conscious red deer.

Studies show that, even within the same family, STT values can be highly variable; also, the methodology is different. There are STT data on conscious, manually restrained European fallow deer (*Dama dama*): An average STT of 18.7 ± 5.1 mm/min and 17.8 ± 3.16 mm/min [8,12]. These values were higher than in manually restrained brown brocket deer (*Mazama gouazoubira*) (8.9 ± 1.8 mm/min) and similar to in sambar deer (*Rusa unicolor*) (STT 18.8 ± 4.7 mm/min), but higher than in anaesthetised Persian fallow deer (*Dama mesopotamica*) (STT 10.5 ± 6.5 mm/min) [12,13,14,15]. In all cases, however, STT was measured for one minute. Our results for conscious red deer are similar to those for European fallow and sambar deer, 18.11 ± 4.33, with a range of 17.33–18.89 in just 30 s. Comparing the results to other species where STT was measured for 30 s, we determined that the STT test values for red deer were among the highest. The STT values were 19.06 ± 3.88 mm/min in horses, and 24.18 ± 6.5 mm/30 s in cows [24,26]. So far, only anaesthetised Persian fallow deer have shown a lower STT, but there are no data on the influence of sedatives on STT—except in horses, where IV xylasine did not have a lowering effect [16]. Also, low doses of α_2_-adrenoceptor agonists, neuroleptics, benzodiazepines, and opioids have no clinically significant effect on aqueous tear production in healthy dogs, but STT-I values increase after intramuscular ketamine. It is shown that higher doses of α_2_-adrenoceptor agonists and combinations of anaesthetics, including inhaled anaesthetics, clinically significantly decrease tear production [17].

The rebound tonometer TonoVet^®^ is widely used in veterinary practice due to its ease of use and simplicity, and it was used to measure intraocular pressure in our study. Study data were obtained using equine calibration settings because they are as close as possible to cervids. These parameters are not known for red deer, so the influence of these settings on the determined IOS values is unknown. However, it should be taken into account that, for setting clinical norms, it is preferable to use the instruments that will be used most often in practice, so that the reference values have clinical meaning [27].

In our study, the IOP values of red deer were no higher than those described for other similar animals (26.33 ± 3.63 mmHg). Intraocular pressure is highly variable in cows (*Bos taurus taurus*); its mean value ± standard deviation is 25.97 ± 14.30 mmHg [28]. Within the deer family, IOP is also varying. In conscious European fallow deer that were manually restrained, IOP was 21.5 ± 5.1 mmHg on the H (horse) setting [12], measured with an applanation tonometer; for sambar deer, it was only 11.4 ± 2.8 mmHg [14]. By comparison, in sedated Persian fallow deer and white-tailed deer (*Odocoileus virginianus*), IOP was 11.9 ± 3.3 mmHg, measured by an applanation tonometer, and 12.87 ± 2.57 mmHg when measured with a rebound tonometer, which is lower than our results [11,13]. Also, in brown brocket deer, IOP was lower when using an applanation tonometer—15.3 ± 3.3 mmHg—but these animals were manually restrained rather than sedated [15]. Anaesthesia has been found to significantly alter both IOP and STT in dogs and cats [29,30], but there are limited data on the effects of different sedatives on IOP in wild animals, especially deer. In laboratory rabbits, a combination of xylasine and ketamine decreased IOP after IV and IM injections [31], but a combination of tiletamine and zolazepam did not cause any decrease in dogs [32]. However, these data were not sufficient to determine the effect on IOP. In veterinary practice, it is important to use reference values according to the type of tonometer, as it has been shown that in some species there are significant differences between the values produced by rebound and applanation tonometers. Villar showed that the mean ± SD IOP values in adult deer were 15.57 ± 2.88 mmHg measured with TonoPen^®^ and 12.87 ± 2.57 with TonoVet^®^. Therefore, TonoPen^®^ significantly overestimated the IOP compared to TonoVet^®^ (*p* < 0.005) [11]. In alpacas, IOP readings were statistically higher with the rebound tonometer compared to the applanation tonometer [33]. Recently, differences have even been shown between two generations of rebound tonometers in horses, dogs, and sheep [34].

These results show that, even within one family, STT and IOP values can be highly variable. Therefore, when evaluating the results of examinations, one should not rely on reference interval values from closely related animals.

The STT and IOP data were obtained during the animals’ annual deworming, so only adult red deer females were used in the study. It was not, therefore, possible to assess the influence of sex on STT and IOP. The influence of both age and sex on these values varies within both species and families; therefore, it would be desirable to continue the research by obtaining data on adult red deer males and young animals of both sexes. Also, the effect of the circadian cycle on STT and IOP in cervids has not been studied. All data were obtained on one day, from 9:00 to 15:00; therefore, changes in the daily cycle, which might influence the test readings, most likely did not affect our IOP results, since the largest differences between the IOP values of different species were observed early in the morning and late evening [35,36,37].

Also, we need to take into account how the size and weight of deer or globe size might influence the results [38]. This is indicated by the variable results within the deer family: The average weight of sambar deer is 102 kg, Asian fallow deer 65 kg, and fallow deer only 18 kg [28]. The red deer is one of the biggest cervids (100–150 kg) [39].

Due to the variability of the IOP and STT measurements associated with certain sedatives in specific species, this study represents the first establishment of normal parameters in conscious, chute-restrained red deer. These parameters can be applied to examinations conducted under similar restraint conditions and allow researchers to perform STT in a shorter period.

## 5. Conclusions

This study provides reference values for STT (17.33–18.89 mm/30 s.) and IOP (26.07–27.55 mmHg) in healthy red deer, showing that STT and IOP evaluating methods, are practical for ophthalmic examination in deer.

## Figures and Tables

**Table 1 animals-14-00940-t001:** Descriptive statistics for STT and IOP measurements in deer (*n* = 60).

Measurements	Mean	±SD	Min	Max	Median	Suggested Reference Interval
Deer						
STT, right eye, mm/min	18.35	±4.54	10.00	28.00	17.00	17.18–19.52
STT, left eye, mm/min	17.87	±4.12	9.00	28.00	18.00	16.80–18.93
STT, both eyes, mm/min	18.11	±4.33	9.00	28.00	18.00	17.33–18.89
IOP, right eye, mmHg	26.96	±4.42	17.00	35.00	27.00	25.81–28.09
IOP, left eye, mmHg	26.67	±3.80	19.00	35.00	26.00	25.70–27.65
IOP, both eyes, mmHg	26.81	±4.11	17.00	35.00	27.00	26.07–27.55

**Table 2 animals-14-00940-t002:** Descriptive statistics for STT and IOP measurements according to the deer age (*n* = 60).

Measurements	Mean	±SD
**Deer age 3–5 years (6 animal)**		
STT, both eyes, mm/min	16.17	±4.12
IOP both eyes, mmHg	28.67	±4.12
**Dear age 6–7 years (34 animal)**		
STT, both eyes, mm/min	18.03	±4.50
IOP, both eyes, mmHg	26.99	±4.05
**Dear 9–11 years (19 animal)**		
STT, both eyes, mm/min	18.81	±4.04
IOP, both eyes, mmHg	25.73	±4.06

## Data Availability

Data are contained within the article.

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
