# Peer review of "Measurement of Tear Production and Intraocular Pressure in Clinically Conscious Normal Captive Red Deer (Cervus elaphus)"

_animals, 2024, doi:10.3390/ani14060940_

Round 1
Reviewer 1 Report
Comments and Suggestions for Authors
The title reflects the content of the work. The study aimed to present intraocular pressure values and measure tear production in a group of healthy and conscious "wild" cervids. The presented publications show that this is one of the first works comparing biometric data of farm fallow deer with wild animals. In this respect, the aim of this study "to present intraocular pressure values and measure tear production in a group of healthy and conscious animals" was achieved. Additionally, as stated in the paper, this is the first attempt to study these parameters in conscious "wild" cervids.
What specific gap in the field does the paper address?
The presented work fits very well into research tracking changes occurring in farm animals due to their use for breeding. Research on deer on farms provides an opportunity to understand the wild animal population better. Deer populations on farms can be treated as a zero sample (complete herd control, optimal nutritional conditions, less impact of unfavourable climatic factors, no stress related to predation, veterinary care, etc.) for comparisons with populations of free-living animals.
The bibliography is extensive and up-to-date. Correct selection of sources: 39 references were used in the work. The items included in the list of references have been selected correctly for the issues discussed.
The measurements were taken methodically correctly. The statistical methods used were correct but insufficient. The authors only presented the type of test used to assess differences between eyeballs. There is no described method for determining reference ranges. The summary is sufficient, without unnecessary generalizations and focuses on valuable points of the manuscript.
Weaknesses of the manuscript:
No transparent methodology for determining reference ranges exists. The reference group should consist of at least 120 healthy individuals to determine the range of reference values. Research is carried out in such a group of subjects, and then, based on the results obtained and using statistical models, the range of reference values is established. If no method is provided for determining the reference range, the conclusion that the study provides reference values is invalid. The only thing that can be said is that the study only provides data to determine reference values.
Detailed notes:
The 3rd paragraph in the "Materials and Methods" chapter is unclear. It shows that 60 individuals were examined, and after randomly examining them, 40 were excluded, so it can be assumed that only 20 individuals were used for further research. Only the "Results" chapter in the first paragraph was explained correctly. Therefore, 1 paragraph from the "Results" chapter should be moved to the "Material and Methods" chapter. After carefully reviewing the submitted work, I declare it suitable for publication in the journal Animals after explaining and presenting the methodology for determining reference ranges.
Author Response
Dear Reviewer,
Thank you so much for your revision and good comments. Please find my comments:
- No transparent methodology for determining reference ranges exists. The reference group should consist of at least 120 healthy individuals to determine the range of reference values. Comment: to calculate the reference range we used the confidence interval method, not the Percentile method, therefore, that shows the lower and upper limits of the means with 95% confidence in the population.
- Research is carried out in such a group of subjects, and then, based on the results obtained and using statistical models, the range of reference values is established. If no method is provided for determining the reference range, the conclusion that the study provides reference values is invalid. The only thing that can be said is that the study only provides data to determine reference values.- Comment: The normal range for STT and IOP was estimated by calculating the confidence interval, i.e. the lower and upper limits of the means with 95% confidence, using SPSS software, added.
3. The 3rd paragraph in the "Materials and Methods" chapter is unclear. It shows that 60 individuals were examined, and after randomly examining them, 40 were excluded, so it can be assumed that only 20 individuals were used for further research. Only the "Results" chapter in the first paragraph was explained correctly. Therefore, 1 paragraph from the "Results" chapter should be moved to the "Material and Methods" chapter. After carefully reviewing the submitted work, I declare it suitable for publication in the journal Animals after explaining and presenting the methodology for determining reference ranges. Comment: Details are corrected in the M&M, sorry for a misunderstanding
Reviewer 2 Report
Comments and Suggestions for Authors
The manuscript "Measurement of Tear Production and intraocular pressure in clinically conscious normal captive red deer does (Cervus elaphus)" describes for the first time the reference values of tear production and intraocular pressure in healthy red deer. Although the results are new and with clinical importance the manuscript needs revision. There are some points which must be improved. I`m sending my suggestions to the Authors:
1. Title of the manuscript: please check the title again carefully. Are you sure that the word "does" at the end of the title is necessary? The same using of this word I found in the abstract and the text of the manuscript.
2. Abstract - see my above comment.
3. Material and methods:
- the number of the permission of the Local Ethics Committee must be added.
- more information about the age of animals is necessary, I suggest to add the table 2 with this details. You wrote only that the animals age varied between 3 and 11 years. Could you subdivided the animals in the smaller groups and compare the results obtained in younger and older red deer (in Table 3).
4. Results - see my above comment.
Author Response
Dear Reviewer,
Thank you so much for your comments.
1. Title of the manuscript: please check the title again carefully. Are you sure that the word "does" at the end of the title is necessary? The same using of this word I found in the abstract and the text of the manuscript. Comment: does are deleted in all manuscript
2. Abstract - see my above comment.Comment: does are deleted in all manuscript
3. Material and methods:
- the number of the permission of the Local Ethics Committee must be added.Comment: added
- more information about the age of animals is necessary, I suggest to add the table 2 with this details. You wrote only that the animals age varied between 3 and 11 years. Could you subdivided the animals in the smaller groups and compare the results obtained in younger and older red deer (in Table 3).Comment: table 2 was made to show values in different age groups
4. Results - see my above comment. Comment: does are deleted in all manuscript
Reviewer 3 Report
Comments and Suggestions for Authors
This study seeks to create reliable reference ranges for the STT and IOP of conscious red deer. Data was collected during annual examinations of adult female red deer as part of routine vaccination and deworming in the spring. Animals were gently restrained for physical examination. An ophthalmic examination was performed. Tear production was evaluated via a thirty second Schirmer tear test, and intraocular pressure was measured using a TonoVet tonometer. Data was gathered for both left and right eyes, and reference values were calculated for left, right, and both eyes.
Overall, the experiment was performed with appropriate scientific rigor and ethical care for the animals. Statistics were performed using standard statistical software. The findings are discussed with the reference values of other related species for additional context. The major findings (the reference ranges) appear valid and are appropriately discussed.
I believe the study is appropriate for publication with minor changes. In the introduction, the authors discuss the importance of ocular exams in general terms, but a greater emphasis on ocular pathology that is either common in deer or specific to deer would be helpful to the reader. If possible, the authors should add further information regarding the prevalence of ocular pathologies in deer and how measuring IOP will help with getting a diagnosis. This information will help convey the necessity of having a reliable IOP reference range as well as the importance of this paper in particular. This could be achieved with a sentence or two, probably on page two near the sentence “Deer are susceptible to a variety of ophthalmic conditions such as congenital, acquired, and infectious diseases that probably increase with age” or in another sensible place in the text.
There is inconsistency with the capitalization of “Red deer” vs “red deer” in the text. Generally, “red” should not be capitalized. “Schirmer” is also misspelled as “Schrimer” within the Simple Summary and Abstract sections. The paper appears well-written overall without any other significant grammatical errors or misspellings.
Author Response
Dear Reviewer,
Thank you for your good comments,
- In the introduction, the authors discuss the importance of ocular exams in general terms, but a greater emphasis on ocular pathology that is either common in deer or specific to deer would be helpful to the reader. If possible, the authors should add further information regarding the prevalence of ocular pathologies in deer and how measuring IOP will help with getting a diagnosis. This information will help convey the necessity of having a reliable IOP reference range as well as the importance of this paper in particular. This could be achieved with a sentence or two, probably on page two near the sentence “Deer are susceptible to a variety of ophthalmic conditions such as congenital, acquired, and infectious diseases that probably increase with age” or in another sensible place in the text. Comment: after this sentence we have info on IOP and STT tests and value of these test during the diagnostics of some diseases.
2. There is inconsistency with the capitalization of “Red deer” vs “red deer” in the text. Generally, “red” should not be capitalized. Comment: corrected
3. “Schirmer” is also misspelled as “Schrimer” within the Simple Summary and Abstract sections. Comment: corrected
Reviewer 4 Report
Comments and Suggestions for Authors
A complete ophthalmological examination is crucial for assessing vision deficits that can significantly impact the quality of life of any living animal. This manuscript presents new reference intervals for tear production and intraocular pressure (IOP) in conscious female Red deer. While the study is well-designed and effectively written, some clarification of the methods and minor efforts in the presentation of results are needed. The discussion thoroughly addresses questions raised by comparisons with previous studies and is supported by a comprehensive literature review. The conclusions are evident, but they need to be further developed in terms of their significance to clinicians and science.

Author Response
Dear Reviewer,
Thank you so much for a great comments.
Round 2
Reviewer 2 Report
Comments and Suggestions for Authors
All the reviewers` suggestions were included within the current version of the paper.